# A System Identification and Implementation of a Soft Sensor for Freeform Bending

**DOI:** 10.3390/ma14164549

**Published:** 2021-08-13

**Authors:** Sophie Charlotte Stebner, Daniel Maier, Ahmed Ismail, Shubham Balyan, Michael Dölz, Boris Lohmann, Wolfram Volk, Sebastian Münstermann

**Affiliations:** 1Integrity of Materials and Structures (IMS), Department of Ferrous Metallurgy (IEHK), RWTH Aachen University, Intzestraße 1, 52072 Aachen, Germany; shubham.balyan@iehk.rwth-aachen.de (S.B.); michael.doelz@iehk.rwth-aachen.de (M.D.); sebastian.muenstermann@iehk.rwth-aachen.de (S.M.); 2Chair of Metal Forming and Casting, Department of Mechanical Engineering, Technical University of Munich, Walther-Meißner-Straße 4, 85748 Garching, Germany; daniel.maier@utg.de (D.M.); wolfram.volk@utg.de (W.V.); 3Chair of Automatic Control, Department of Mechanical Engineering, Technical University of Munich, Boltzmannstraße 15, 85748 Garching, Germany; a.ismail@tum.de (A.I.); lohmann@tum.de (B.L.)

**Keywords:** soft sensor, extended Kalman filter, freeform bending, residual hoop stresses, hardness, control-loop, measurement suitability, ultrasonic contact impedance

## Abstract

The primary goal of this study is the formulation of a soft sensor that predicts industrially relevant mechanical properties for freeform bending. This serves as the foundation of a closed-loop property control. It is hypothesized that by inline measurement of hardness, predictions regarding residual hoop stresses, local strength and strain level can be achieved. A novel hardness-based correlation scheme is introduced, which is implemented into an extended Kalman filter (EKF) and allows an inline prediction of local strength, residual hoop stresses and plasticity. Furthermore, the ultrasonic contact impedance (UCI) method is validated as a suitable inline measuring solution.

## 1. Introduction

Industrial plants are usually equipped with a manifold of sensors that are used to monitor as well as to control processes. Dependent upon the monitored process, the equipped sensors may have sampling rates that are too low or the measurements can only be conducted through extensive offline analyses, which is unsuitable for efficient process control [1,2]. Thus, researchers started developing predictive mathematical models from physically measured data sets that allow the continuous monitoring of relevant state variables of the processes. These predictive models, in combination with physically measured sensor data, are so-called soft sensors [1,2]. Soft sensors find their application in various fields, such as the control and optimization of bioprocesses [3], quality monitoring in the petrochemical industry [4] or even in the field of soft robotics [5]. Dependent upon the physical content of the soft sensor, the soft sensor model can be differentiated into two categories, namely model-driven and data-driven soft sensors. Model-driven soft sensors, also called white-box models, completely rely on physical-mathematical models of the process. Data-driven soft sensors, also called black-box models, are empirical models that purely rely on statistical relationships. A combination of these two models is also common, and is referred to as a grey-box model [1,2,6]. The authors in [6] present an overview of the characterization and classification of models and modelling specifically in metal forming, as well as in introduction to model selection.

The freeform bending process with a movable die offers an approach to bending complex geometries without changing the bending tool. Currently, a set geometry is bent while neglecting the mechanical properties of the workpiece. As freeform bending causes plastic deformation of the tube, not only does the material harden but the residual stresses in the part are influenced significantly [7]. Yet, the residual stress state, the hardening of the material and the ductility influences further processing steps down to the production line as well as the service behavior of the component [8,9,10]. Thus, the need to control the mechanical properties decoupled from a set geometry arises, as this allows precise adjustment of the mechanical properties. The authors in [7] describe the bending process in detail and study the influence that different degrees of freedom of the freeform bending process have on the mechanical properties. They introduced a novel bending strategy, so-called non-tangential bending, which now allows the mechanical properties to be influenced while keeping a set geometry. Compared to normal freeform bending, the bending die has a non-tangential position during maximum deflection to the bent component. This position can lead to either underbending or overbending of the tube, see Figure 1. Non-tangential bending extends the result space for freeform bending and can be used for a decoupling of bent geometry (radius and angle of the tube) and the resulting properties of the tube [7].

The authors in [7] namely introduce seven bending strategies (9 mm 22 deg, 10 mm 16 deg, 10 mm 20 deg, 11 mm 13 deg, 11 mm 14 deg, 12 mm 12 deg and 12 mm 13 deg), where the denotation of, e.g., 9 mm 22 deg means that the bending tool was deflected by 9 mm and had a rotation angle of 22 deg. These bending strategies offer the same set geometry and constitute the tubes studied in this paper.

As the mechanical properties can currently only be determined by time-consuming, offline measurements, an intelligent solution in the form of a soft sensor is needed, as this allows inline prediction of inline unmeasurable quantities and can then serve as a basis for a control loop based on mechanical properties.

To develop a soft sensor for the freeform bending process with movable die, an analysis of the process data delivered by the physically measured data sets is necessary. This allows the derivation of a concept for correlating state variables in the soft sensor. This paper focuses on the process data analysis and the measurement equipment suitability and proposes a novel correlation scheme for local strength, strain level and residual hoop stresses derived from ultrasonic contact impedance (UCI) measurements for freeform bending, which will be introduced in Section 2. Part of the process data analysis is how the novel freeform bending strategy of non-tangential bending influences the mechanical properties of the material. Due to the bending process, the material will undergo plastic deformation, which will affect the residual stresses, plasticity and local strength [8,11].

Residual stresses are multi-axial stresses in equilibrium within a closed system that is not subject to any external forces or torques [10]. The residual stress states in the workpiece are influenced by nearly all manufacturing processes. A change in residual stress state can be caused by, e.g., locally varying thermal conditions or inhomogeneous plastic deformation [9,12]. In most cases, residual stress states can be neglected, yet in some cases the residual stress state must be considered as it determines the technical application of the workpiece. If not considered, the residual stress state can lead to failure of the workpiece such as crack formation, distortion, or maybe even detrimental failure [9,10]. Residual stresses are categorized into three types, namely residual stresses of type I, II and III. They can be of tensile and compressive nature and are dependent upon the orientation in the tube. They can be differentiated into hoop residual stresses, longitudinal residual stresses and radial residual stresses [13]. Tensile residual stresses can render an entire workpiece useless if the component must withstand high load stresses, while compressive residual stress states can positively influence the workpiece’s lifespan [12].

The residual stress state can currently only be quantitatively investigated by extensive offline analyses such as (semi-) destructive investigation methods, e.g., the drill hole method [14,15] and sectioning methods [16], as well as non-destructive investigation methods such as the x-ray diffraction measurement or magnetic methods, e.g., Barkhausen noise analysis [17,18]. Magnetic Barkhausen noise, in particular, poses a good alternative, but the signal among different types of materials is not comparable as different microstructures lead to different micromagnetic signals [18]. Yet, to make freeform bending more efficient, an inline monitoring of the residual stresses is needed that can also be used across a variety of materials, which will be investigated in this work.

In addition to the residual stress state, the strain hardening of the material, due to the bending process, needs to be investigated. Strain hardening of a material is influenced namely by the degree of deformation ϕ, the strain rate $\dot{\varepsilon }$ and the Temperature T. The authors in [7] have shown that the influence of the strain rate $\dot{\varepsilon }$ on strain hardening during freeform bending with a movable die can be neglected. During forming, ϕ changes the order and density of the dislocations, phase distribution and grain shape of the material. During cold forming, an increased ϕ causes the dislocation density of the material to rise, meaning the material strain hardens while the elongation capacity decreases [8,11]. Depending on the loading case of the bent workpiece, it is necessary to know how much the material strain hardened and elongated during the bending process, as it will influence the application and lifespan of the workpiece [18]. Thus, the local strength as well as plasticity are of importance to monitor during the bending process.

It is commonly known that tensile testing provides information on materials’ characteristics such as yield strength, tensile strength, strain hardening behavior and maximum elongation [19]. It is also commonly known that the tensile strength of a material can be analytically correlated to the measured hardness [20]. Furthermore, an influence of residual stresses on hardness have also been proven [21]. Yet, an analytic correlation between hardness, the local strength and residual stresses has not yet been proposed. Thus, this work proposes a novel correlation scheme based on measured hardness, Section 2.3, that is implemented into a soft sensor.

For the soft sensor itself, a suitable model must be identified. When analyzing the system closely, the development of the mechanical properties influenced by the freeform bending process depicts a dynamic, nonlinear system. The term of a dynamic system describes a state space within which the defined coordinates describe the state at any given point in combination with a dynamical rule using the present values of the state variables in order to describe the state variables’ direct future [22]. While nonlinearity of a dynamic system means that the system’s output cannot be described by a linear operator applied to the system’s input [23]. The nonlinearity of the system is derived from the results of [7], as the residual hoop stress state influences the hardness nonlinearily. Models for nonlinear, dynamic systems are manifold. Among these are for example NARMAX models [24], neural networks relying solely on statistical relationships between variables [24], Kalman-filters [24] and various others. The authors have chosen the extended Kalman filter to model the development of the mechanical properties, as this is a well-researched algorithm for non-linear dynamic systems that has a relatively low complexity [25]. As no previous filter has been introduced for the freeform bending process, the authors chose the EKF as the basis for the proposed soft sensor. Thus, any other model will not be discussed further.

The extended Kalman filter (EKF) is an advancement of the Kalman filter. As the development of the mechanical properties depicts a nonlinear system, the EKF is needed. The EKF linearizes the system about a current estimate in each time step by implementing a Jacobian matrix, meaning a first-order partial derivative of a vector function with respect to a vector [25]. To give a state estimate, the algorithm utilizes two steps: a prediction and an update step that are modelled by the following equations [25]:


*Prediction Step*
(1)$$\mathrm{State}\;\mathrm{Estimate}\;{\hat{x}}_{k}^{-}=g\left({\hat{x}}_{k-1}^{+},\;u_{k-1}\right)$$
(2)$$\mathrm{Error}\;\mathrm{Covariance}\;P_{k}^{-}=F_{k-1}P_{k-1}^{+}F_{k-1}^{T}+Q$$



*Update Step*
(3)$$\mathrm{Measurement}\;\mathrm{Residual}\;{\tilde{y}}_{k}=z_{k}-h\left({\hat{x}}_{k}^{-}\right)$$
(4)$$\mathrm{Kalman}\;\mathrm{Gain}\;K_{k}=P_{k}^{-}H_{k}^{T}\;{\left(R+H_{k}P_{k}^{-}H_{k}^{T}\right)}^{-1}$$
(5)$$\mathrm{Updated}\;\mathrm{State}\;\mathrm{Estimate}\;{\hat{x}}_{k}^{+}={\hat{x}}_{k}^{-}+K_{k}\tilde{y}$$
(6)$$\mathrm{Updated}\;\mathrm{Error}\;\mathrm{Covariance}\;P_{k}^{+}=\left(I-K_{k}H_{k}\right)P_{k}^{-}$$


The hat operator describes that the variable is an estimate, whereas the superscripts − and + denote that the estimates are predicted or updated estimates of the variable. Term P is called the state error covariance and describes the error covariance the filter thinks the estimate error has. Due to its summation with Q, the error covariance increases in the prediction step, which means that the filter is more uncertain in the estimate of the state variable after prediction [25]. During the update step, the measurement residual, ${\tilde{y}}_{k}$, is computed first. It is the difference between the true measurement, $z_{k}$, and the estimated measurement, $h\left({\hat{x}}_{k}^{-}\right)$, where h is the measurement matrix. The residual is later multiplied by the Kalman gain $K_{k}$ to provide the updated state estimate ${\hat{x}}_{k}^{+}$. The filter then computes the updated error covariance $P_{k}^{+}$ to use in the next time step. The updated error covariance is smaller than the predicted error covariance, meaning the filter became more certain. F and H are two Jacobian matrices, which serve the purpose of linearizing the nonlinear system [25].

The following sections will present how the correlation scheme based on hardness deriving from local strength, ductility and residual hoop stresses is formulated and how each term influences the hardness. Furthermore, the measurement suitability of UCI hardness measurements is analyzed as an inline-monitoring solution by investigation surface influences as well as cross-referencing the UCI method with Vickers hardness scale HV 10 measurements (Dia-Testor 731, Instron Wolpert, Ludwigshafen, Germany). The derived correlation scheme and all influencing parameters are lastly implemented into the EKF to enable inline monitoring of all relevant mechanical properties during the freeform bending process.

## 2. Materials and Methods

This section gives a short overview of the investigated material. Furthermore, inline measurement suitability of UCI hardness measurement equipment is investigated. The focus lies on the introduction of the novel correlation scheme based on UCI hardness measurements that depict the basis for inline monitoring of the residual hoop stresses, the local strength, as well as the plasticity during freeform bending.

### 2.1. P235 TR1

The material used in this investigation is P235 TR1. It is a low-alloyed steel grade that is designed for pressure purposes, where the alphanumerical definition of TR1 depicts a grade without fixed values for the impact energy or a defined aluminum content [26,27]. The material is ferritic with a small fraction of perlite, see Figure 2. The chemical composition investigated by optical emission spectrometer can be taken from Table 1.

The circular steel tubes have an outer diameter of 42.4 ± 0.5 mm, a thickness of 2.6 ± 0.3 mm and a length of 800 ± 2 mm. The tubes are welded longitudinally and are finished evenly both on the inner and outer part of the tube as is necessary for freeform bending. The tubes are manufactured according to DIN EN 10217-1 and DIN EN 10219-1 [26,27].

### 2.2. Inline-Measurement Suitability and Measurement Uncertainties of UCI-Hardness Testing

Firstly, it must be established whether UCI hardness testing is suitable for inline measuring of hardness. UCI hardness testing uses a longitudinally oscillating rod with a Vickers diamond at the end. This diamond is pushed into the object to be tested. The defined load is usually applied by a spring. The rod oscillates at its natural resonant frequency, which depends mainly on its length. When the Vickers diamond penetrates the sample, this vibration is damped. The damping of the rod and thus the change in frequency to be measured depends on the size of the contact area between diamond and sample. In combination with the known load and the stored calibration values, the measured frequency directly determines the hardness of the material [18,28]. The advantage of the UCI hardness measuring method is that it offers a good inline-measuring capability as the measuring equipment is portable, can be integrated into the freeform bending machine, and is of a semi-destructive nature. However, as faulty test equipment and measuring uncertainties can lead to faulty test results, the UCI hardness measurements will be investigated regarding their inline measurement suitability as well as regarding the systematic and statistical measuring uncertainties.

The measurement suitability will be examined by cross-referencing UCI hardness measurements with HV 10 measurements. The HV 10 measuring method also utilizes a diamond attached to a straight pyramid with an angle of 136° between the opposing sites to indent the test specimen surface. The diamond is pushed vertically into the material with a defined test force F. The Vickers hardness HV 10 is then calculated by measuring the mean value of the diagonals that result from the indentation of the diamond and are corrected by a factor due to the curvature in the tube [29]. To conduct the HV 10 measurements, the surface of the tube must be polished, because HV 10 relies on the optical measurement of the indentation. A tarnished surface cannot be evaluated. Thus, the surface to be investigated was treated by a fan grinder with the grit of 80 to clean off any tarnishes. HV 10 measurements were chosen as the results are comparable to UCI hardness measurements and it is an established method to determine hardness in a stationary fashion [18].

A method used in statistics to compare two measurement methods of the same variable that serves as a tool to analyze the concordance between these two methods is the so-called Bland–Altman plot. Such a plot will be defined for UCI hardness measurements and HV 10 measurements. Given that the difference between the measurements is normally distributed, plotting the difference between the measured values of UCI hardness and HV 10 measurements against the mean value and implementing three lines that define the limits of agreement, i.e., the mean value of the difference as well as the mean value of the difference plus or minus 1.96 times the standard deviation of the difference, it is possible to determine the agreement between the two measuring tools. It is recommended that 95% of the data points should lie within the limits of agreement. If that is the case, HV 10 measurements can be replaced by UCI hardness measurements [30,31]. To test the normal distribution of the difference between the measurements, a Kolmogorov–Smirnov test was conducted [32].

Additionally, a study regarding influences of the surface condition of the results is conducted, as [33] have shown that surface roughness can influence measurements. Surface roughness analysis was performed by investigating two different surfaces—the untreated and polished surface. The polished surface was treated by a fan grinder with a grit of 80 along the inside and outside surfaces of the material. Denotation of inside and outside is used according to Figure 1.

Lastly, measurement uncertainty is analyzed by determining the systematic measurement error and statistical measurement uncertainty of the UCI hardness measuring device (INNOVATEST MET-U1A, HaBu Hauck Prüftechnik GmbH, Hochdorf-Assenheim, Germany). Systematic measurement errors are deviations in the measurement results that are the same every time a measurement is taken. They cannot be identified by statistical analyses. This type of error is difficult to determine as well as to eliminate. Usually, systematic errors are due to the experimental set up or the calibration of the equipment [34]. Statistical measurement uncertainty is a characteristic value obtained from measurements and is used together with the measurement result to identify a range of values for the true value of the measurand [34]. A simple method to determine statistical measurement uncertainty is determining the repeatability standard deviation as follows:(7)$$\mathrm{Repeatability}\;\mathrm{standard}\;\mathrm{deviation}\;s=\sqrt{\frac{1}{n-1}\sum_{i=1}^{n}{\left(x_{i}-\bar{x}\right)}^{2}}$$
where $x_{i}$ is the corresponding single measured value and $\bar{x}$ the arithmetic mean of all measurements. In general, the larger the statistic, the better the probability distribution of the true value can be estimated from the standard deviation and mean [34]. Thus, a statistic of thirty measurements under equal conditions were conducted.

### 2.3. Correlation Scheme Based on Hardness

The core of this work lies in the proposed, novel correlation scheme based on hardness that offers an analytical relationship between hardness, local strength and residual stresses within the tube (see Formula (8)).
(8)$$\mathrm{Correlation}\;\;\;\mathrm{Scheme}\;\;\;HV_{Total}=HV_{Groundstate}+HV_{Strain\;Hardening}+HV_{Residual\;Stresses}$$

The correlation scheme hypothesizes that the measured hardness of the total system is influenced especially by three parameters:The hardness of the ground state, meaning the hardness of the material itself without any influences of strain hardening or residual stresses due to shaping within the material. It can be measured by hardness measurements taken from tensile test samples, as these are in a uniaxial stress state.An increase in hardness due to the strain hardening of the material because of plastic deformation by bending.Lastly, the residual stresses within the steel tube.

As the algorithm derives the local strength from the hardness, the information of the strength will then be utilized by the soft sensor to derive the level of plasticity.

Each of the above mentioned parameters of the correlation scheme as well as the derivation of the level of plasticity are investigated closely within this work. The tests were performed on non-tangentially bent steel tubes according to [7]. All influencing parameters are investigated separately. The next section introduces the methods of determining the individual parameters.

#### 2.3.1. Determination of *HV_Total_*

From the total hardness measurements, the soft sensor will ultimately derive the local strength, residual hoop stresses and plasticity level. Total hardness was determined by conducting UCI hardness measurements by hand along the surface of the steel tubes (see Figure 3). Three hardness measurements were taken every two centimeters apart on the inside as well as the outside of the tube. Only three measurements were taken, as the measurements need to have a small distance between them so as not to measure the same point twice, since this leads to faulty results and can result in the indenter breaking.

#### 2.3.2. Determination of *HV_Groundstate_*, *HV_Strain Hardening_* and Plasticity

In conducting tensile tests according to DIN EN ISO 6892-1 [18] with miniature tensile test samples, both the strain hardening and plasticity level of the material can be studied. By taking the hardness value of miniature tensile test samples, an overview of the hardness without any influences due to shaping the material into the tube is given. Ten specimens were tensile tested until failure to determine the strain hardening behavior for an array of samples. These data were then used to model the tensile test with Abaqus CAE in order to derive the local strength and ultimately the true plasticity level. A flow curve was fitted using the Ludwik–Voce Equation (see Formula (9)). The according fitting parameters can be taken from Table 2.
(9)$$\mathrm{Ludwik}\text{-}\mathrm{Voce}\;\;\;\mathrm{Equation}\;\;\;\bar{\sigma }\left({\bar{\varepsilon }}_{p}\right)=\alpha_{L\;}\left(b_{L}+A_{L}\cdot {\bar{\varepsilon }}_{p}{}^{n_{L}}\right)+\left(1-\alpha_{L\;}\right)\left(k_{0,V}+Q_{V}\cdot \left[1-e^{-\beta_{V}\cdot {\bar{\varepsilon }}_{p}}\right]\right)$$

The individual fitting parameters were set to the following values:

Another fourteen specimens were tensile tested and stopped at certain plasticity points, namely 1% until 14% of conventional plastic strain, to measure the hardness the material has at various plasticity points. By analyzing the displacement in the stopped tensile test, the according strength values can be derived from the simulation, thus the strain hardening behavior can be determined.

After testing, the stopped miniature tensile test samples were embedded into epoxy resin and UCI measurements were taken along the surface of the reduced section to determine $HV_{Groundstate}$ and $HV_{Strain\;Hardening}$. In taking the UCI hardness and matching the hardness to the local strength value, the relationship between hardness and local strength can be derived and subsequently implemented into the soft sensor.

The level of plasticity was also derived by using the data from the simulation. In analyzing the plasticity level and varying strength values derived from hardness measurements by the EKF, the strength values can be cross-referenced with the tensile tests, thus offering insights into the plasticity level reached during plastic deformation, see Section 2.3.4.

#### 2.3.3. Determination of *HV_Residual Stresses_*

To determine the last term of the correlation scheme, $HV_{Residual\;Stresses}$, the influence that hoop stress has on hardness was closely analyzed. To do so, the approach the authors in [7,16] chose was utilized and this analysis relies on their results.

In plotting the residual stresses against the hardness, a regression can be derived to model the development of the residual hoop stresses with varying hardness of the EKF. The regression results that are implemented into the EKF will be presented in Section 3.5 and Section 3.6.

#### 2.3.4. Implementation into the Soft Sensor

The previously determined relationships now need to be implemented into the soft sensor. The soft sensor computes the different mechanical properties in two steps. Firstly, the EKF is implemented, which derives and predicts the local strength and residual hoop stresses. In the second step, after the EKF derives the local strength value, the soft sensor then uses the strength value and scans the tensile test data for the same strength values. The soft sensor starts to scan the data when plastic deformation begins as the elastic deformation behavior is not of interest and could lead to faulty predictions. After the tensile test data is scanned, the soft sensor then returns the true strain that was induced during bending.

For the EKF, an initial state needs to be declared. In this case, the initial state is a vector that describes the hardness, residual hoop stresses and strength of the material. For the initial state, uncertainty in form of the covariance matrix needs to be implemented, namely P in Formula (2). As the initial state was a vector, the covariance matrix is now a matrix containing the implemented three states. The values for Q in Formula (2) depend on the sensor accuracy. If the sensor accuracy is high, smaller values should be initialized, whereas larger values should be implemented if the sensor accuracy is low. In initialization, the values were set to 0.9. Fundamental to the algorithm is the dynamics matrix and the Jacobian matrix thereof. The dynamics matrix depicts how the residual hoop stresses and local strength values develop depending on the hardness. This means that the analytical relationships that were determined in Section 2.3.2 and Section 2.3.3 are implemented into the dynamics matrix and determine the state estimate ${\hat{x}}_{k}^{-}$ in Formula (1). The Jacobian of the dynamics matrix, namely F from Formula (2), is then used to calculate the error covariance and influences the Kalman gain. The EKF must also be given what is measured and how the measurement relates to the initialized state vector. Again, the filter is told that measurement of hardness is computed to strength and residual hoop stresses. Then, the Jacobian of the measurement matrix with respect to the state matrix is computed and is utilized in the computing of the measurement residual, Formula (3), of the update step. The filter now approximates the current measurement by multiplying the predicted state with the measurement matrix, Formula (4). In multiplying the residual and the Kalman gain, the correction of the predicted term based on the new measurements is obtained, Formula (5). When the updated state estimate is computed, the error covariance is computed (Formula (6)) and used in the next prediction step. The subscript k describes in this case the spatial progression along the steel tube.

When in use, the algorithm reads a csv-file with UCI hardness measurements taken previously on the steel tube. The data is then processed by the EKF, meaning that the residual hoop stresses and strength are derived and their development is predicted. The predicted local strength is then used to derive the level of plasticity. This means that the EKF returns a local strength value and this term is then used to scan the tensile test data for a matching pair. As already mentioned, the scan begins with plastic deformation to circumvent any mismatches with strength matches during elastic deformation. Ultimately, the algorithm provides estimates and predictions for residual hoop stresses, local strength and plasticity based on hardness measurements by utilizing mathematically derived relationships. The operating principle of the soft sensor is illustrated in Algorithm 1 operating Principle of Soft sensor. The EKF predicts strength and residual hoop stresses; the soft sensor then uses the predicted strength values from the EKF named *predList.Strength* and scans the tensile test data to return the plasticity level in *plasticityList*.

**Algorithm 1**: Extended Kalman Filter for Material Parameter Prediction **Input**: Hardness values on inside/outside at different points on the material
 **Output**: Intermediate output: Predicted strength and hoop stress values for the corresponding hardness values
    Final output: Plasticity range for each of the predicted strength values
1 *initialize empty material parameter prediction list ’predList’ and ’plasticityList’*2 *initialize a pandas dataframe (distance and inside-outside hardness)*3 *compute ‘strength’ values for the corresponding hardness values using the correlation: **strength = hardness × 2.88***4 *compute ‘hoop stress’ values for the corresponding hardness values using the correlation: **hoop-stress = 46.2502 × exp(0.00992 × hardness)***5 *initialize x_0_ (initial state vector). If nothing is known, simply enter zero here.*6 *define a state matrix, which consists of all possible attributes in the system and a dynamics matrix with respect to the state matrix.*7 *compute the Jacobian matrix J_A_ (first-order partial derivatives) of the dynamics matrix with respect to the state matrix.*8 *define a measurement matrix which consists of all attributes that are being measured in the system and compute its Jacobian matrix J_H_ with respect to the state matrix.*9 *the non-linear *g*(*x*_*k*_, *u*) consists of all the regression expressions, where the control input u is zero and function h(x) is the measuring function that consists of the measurements from regression.*10 *initialize state co-variance matrix **P** to add uncertainty to the initial state (if the sensors are not accurate) and process noise co-variance matrix **Q** for the ‘prediction’ step.*11 *initialize measurement noise co-variance matrix **R** (use smaller values if sensors are accurate) **for the** ‘correction’ step.*12 **Function** Prediction (·):13 Project the state ahead: $x_{k+1}=g\left(x_{k},\;u\right)=\left[\begin{array}{c} 2.88\times hardness\\ 46.2502\;e^{0.00992\times hardness}\\ hardness \end{array}\right]\;$14 Project the error co-variance ahead: $P_{k+1}=J_{k}\;P_{k}\;J_{A}^{T}+Q$15 **return**$x_{k+1}$16 **Function** Correction (z):17 Compute the Kalman Gain: $K_{k}=P_{k}\;J_{H}^{T}{\left(J_{H}\;P_{k}\;J_{H}^{T}+R\right)}^{-1}$18 Define the measuring function: $h\left(x_{k}\right)=\left[\begin{array}{c} strength\\ hoop-stress \end{array}\right]=\left[\begin{array}{c} 2.88\times hardness\\ \;46.2502\;e^{0.00992\times hardness} \end{array}\;\right]$19 Update the estimate via measurement: $x_{k}=x_{k}+K_{k}\left(z_{k}-h\left(x_{k}\right)\right)$20 Update error co-variance: $P_{k}=\left(I-K_{k}\;J_{H}\right)\;P_{k}$
21 **for***z in measured values***do**: 22   *val = Prediction ()*23   Add *val* to *predList*24   *Correction (z)*25 **end**26 ***Computation of Plasticity after EKF predicted strength***27 *initialize a pandas dataframe ’df’ (measured strength and plasticity values)*28 *perform the required rounding off for the measured as well as the predicted strength values to get an appropriate match.*29 **for** value *in predList.Strength*
**do**30   *result = df.loc[df[‘Strength’] == value].Plasticity*31   Add *result* to *plasticityList*32 **end**33 **return***plasticityList*


## 3. Results and Discussion

This section presents the results and the discussion of this work. Firstly, the measuring equipment suitability is analyzed where UCI hardness measurements in comparison to HV 10 measurements are shown. Additionally, surface roughness influences are given. Yet, the focus lies in the derivation of local strength, residual hoop stresses, and plasticity by hardness measurements. The results for each term in the correlation scheme will be presented and a regression drawn. Furthermore, the implementation of the regressions in the EKF algorithm and the predictions generated by the soft sensor will be introduced.

### 3.1. Measurement Equipment Suitability—Comparison of Measuring Methods

The measurement suitability of the UCI hardness measurements is investigated by comparing the UCI hardness values with HV 10 measurements. This was done by performing both UCI hardness measurements and HV 10 measurements on all seven differently non-tangentially bent steel tubes. Figure 4 shows exemplarily the comparison of UCI hardness vs. HV 10 hardness on tubes 10mm16deg both on the inside and the outside of the tube. Based on that data, a Bland–Altman plot was formulated for the UCI hardness and HV 10 measurements to ultimately prove whether UCI hardness measurements are suitable for inline measuring.

In Figure 4, the UCI measurements scatter more than the HV 10 measurements. However, it is visible that both measurements suggest the same hardness development along the tube. Considering that UCI hardness measurements are taken by hand, a higher scatter in measurement data is plausible. Even though the steel tube was clamped tightly during measurement and the measurements were conducted by the same person, manual measurements have the tendency to scatter in a larger fashion.

To quantitatively describe whether UCI hardness measurements are suitable for inline measuring, a Bland–Altman plot was formulated according to Section 2.2. The Kolmogorov–Smirnov test shows that for a confidence interval of α = 0.05 the critical value of 0.2342 is not surpassed by the maximum deviation in the test statistic, namely 0.0808, meaning the null hypothesis will not be rejected. Thus, the difference is distributed normally, and a Bland–Altman plot can be formulated to compare UCI hardness and HV 10 measurements. The Bland–Altman plot shows that the mean of the difference is −0.733 and the limits of agreement are −0.733 + 1.96 × SD and −0.733 − 1.96 × SD. All data points lie within the defined limits of agreement, meaning that UCI hardness measurements offer a good alternative to HV 10 measurements and are suitable as an inline measuring method [30,31].

### 3.2. Measurement Equipment Suitability—Investigation of Surface Roughness Influences

Figure 5 shows the development of hardness measurements along the inside and outside of the steel tube with treated and untreated surfaces. All seven non-tangentially bent steel tubes were examined, but only two exemplary tubes are shown in the following figures, as the results are comparable.

Surface roughness influences, seen in Figure 5, do exist for the UCI hardness measurements. Outliers become less prominent in the treated surfaces, yet both measurements describe identical trends in hardness. Surface roughness in this case means a tarnished surface and roughness due to lubricants from bending. Although an inline surface preparation will not be possible during the bending process, the authors decided to formulate the correlation scheme based on measurements with treated surfaces, particularly as the individual correlation terms of $HV_{Strain\;Hardening}$ and $HV_{Residual\;Stress\;}$ were also determined on samples with a treated surface. It is permissible to formulate the correlation scheme based on data with treated surfaces, as the surface roughness does not have an influence on the underlying mechanical properties such as strain hardening and residual stress state variations. Furthermore, the UCI measurement method relies on the dampening of the oscillation in the rod and not on an optical evaluation. Thus, it is acceptable to perform measurements on surfaces with a higher roughness later on [33].

### 3.3. Measurement Equipment Suitability–Investigation of Systematic Measurement Error and Statistical Measurement Uncertainty

As a systematic measurement error, the authors could identify that the UCI measurements along the surface were taken by hand. To minimize the measurement error, all hand measurements were taken by the same person and a custom-made guider for the UCI hardness measuring device was utilized. All steel tubes were tightly clamped during the measurements as to eliminate any influences due to displacement of the sample. As the UCI hardness measurement equipment will ultimately be incorporated into the machine and guided by a machine, the systematic measurement error “manual testing” will be eliminated.

The statistical measurement uncertainty was determined by measuring a treated, unbent steel tube in an area of two centimeters with thirty measurements in order to derive the standard deviation. The following Table 3 depicts the statistical data as well as the standard deviation determined according to Formula (7).

Thus, yielding a statistical uncertainty determined by a standard deviation of 6.5 HV 1.

In conclusion, a new inline measuring sensor for freeform bending could be established.

### 3.4. HV_Total_

*HV_Total_* development along the non-tangentially bent and treated surface steel tubes can be seen in Figure 6a,b, while Figure 6c,d give an exemplary overview of the hardness development of non-tangentially bent steel tube 10mm16deg. Error bars derived from the standard deviation are also plotted in the diagram.

As expected, the hardness seen in Figure 6 increases slightly where the die is fully deflected, meaning where ϕ is largest, for all non-tangentially bent tubes. Although the measurements scatter, the general trend can be seen. HV 1 measurements lie for inside measurements within an interval of 147 HV 1 to 192 HV 1 and for the outside measurements within an interval of 137 HV 1 and 198 HV 1. It can also be seen that the hardness does not increase significantly for any of the non-tangentially bent tubes, which is also suggested by the conventional stress–strain curves seen in Figure 7, showing that the material does not have a pronounced strain hardening behavior.

### 3.5. HV_Groundstate_, HV_Strain Hardening_ and Plasticity

The results of the tensile tests can be taken from Figure 7. Depicted are the conventional stress–strain curves for the tensile tests conducted at room temperature as well as the numerically modelled tensile test. The conventional strains at fracture vary from 0.19 to 0.26. The tensile test shows that the material does not strain harden strongly. Figure 7 again shows that the material properties scatter strongly. Sample 10 is an extreme outlier and has a tensile strength value lower by 100 MPa compared to the rest of the samples. The strong scatter in the material properties is permissible according to the norm [26,27], yet also results in difficulty of precise regression formulation. It is important to emphasize that investigations, especially regarding the fluctuations in the base material, need to be conducted and accounted for in the algorithm.

Using this data and taking hardness measurements in the different conventional plastic strain stages, a relationship between hardness and local strength can be derived in Figure 8. The hardness and local strength are plotted against the conventional strain level.

It can be seen that the local strength value does not change by much as the interval lies between 465 MPa and 558 MPa even after the conventional plastic strain level rises up to 10%. Compared to this, the hardness values scatter in a higher fashion on an interval between 160 HV 1 and 210 HV 1. However, it is noticeable that hardness and local strength differ from each other by a scaling factor. When local strength and hardness are put into proportion, hardness scaled up by a factor of 2.88 on average results in the strength value. This scaling factor is implemented into the soft sensor to inline monitor the local strength.

To assume the relationship between hardness and local strength to be a scaling factor is suitable for a primary approach, as the material does not have a pronounced strain hardening behavior, elastic material behavior is not of interest and it is a well-known fact that base hardness can be reinterpreted as the tensile strength [20]. Further investigations should be conducted to generate more data. A regression between hardness and local strength should be derived using a material that has a pronounced strain hardening behavior.

To derive the true strain level introduced during bending, the numerical model was utilized. The stopped tensile tests at various conventional plastic strains, 1% to 10%, were analyzed regarding the displacement. The displacement points were compared with the simulation to gain insight into the true strain introduced by bending, as the conventional strain does not describe the actual strains in the critical areas because it does not consider the change in cross-section. The following Figure 9, shows the conventional strain values plotted against the equivalent plastic strain (PEEQ) values from the simulation. The true strain values from the simulation were then used by the algorithm to predict the true strain level in the bent tube. Figure 9 shows that the true strain increases exponentially compared to the conventional strains, as expected.

It must be noted that only plasticity levels up to 10% conventional plastic strain are included. This is due to the fact that beginning at 11% conventional plastic strain, the test samples start to neck diffusely, meaning necking both in the width and thickness direction occurs [35]. Thus, a uniaxial stress state no longer exists and a multiaxial stress state will be present in the material [8]. To still monitor the local strength and true strains but without any influences due to a change in stress state, only the data before necking is used. This is also permissible, as the bent steel tubes do not neck during bending.

To further improve the filter, tensile tests utilizing Zwicks optical measurement equipment Aramis are anticipated. These will provide meticulous data on the local true strain and can be used to derive a regression between plasticity and hardness directly, which can be implemented into the dynamics matrix of the filter.

### 3.6. HV_Residual Stresses_

As mentioned in Section 2.3.3, the regression between residual hoop stresses and hardness relies on the results in [7]. The chosen regression to inline monitor the residual hoop stresses can be seen in Figure 10.

The regression function selected is:(10)$$\mathrm{Regression}\;\;\;\mathrm{Function}\;\;\;y\left(x\right)=46.2502\exp0.00992x$$

This fit offers a coefficient of determination of 0.8555.

The modelling of the residual hoop stresses as an exponential function is also permissible for implementation into the algorithm. The residual of R² = 0.8555 suggests that the function represents the development adequately. Although a quadratic fit would have led to a higher residual of the regression, the development of the residual stresses was modelled with an exponential fit. The quadratic fit would have suggested that at lower hardness the residual hoop stresses would increase again. That is not the trend shown in [7]. Further data points will be generated for proper validation of the primary exponential fit. Additionally, investigations regarding residual stress influences on hardness need to be conducted as currently only residual hoop stresses are monitored. Longitudinal residual stresses and radial residual stresses should also be incorporated as they also determine the application of the work piece.

### 3.7. Implementation of Regressions into Soft Sensor

After implementing the regressions into the dynamics matrix and defining the Jacobian thereof, the soft sensor derives local strength and residual hoop stresses as seen in Figure 11a,b. For the computation of the algorithm, the algorithm utilizes the hardness values without standard deviation. According to the Kolmogorov-Smirnoff test, the test statistics for measurement uncertainty is normally distributed for α=0.05. For the taken test statistic, the true value lies for 95% of all samples in the confidence interval of [188.81; 193.45]. Thus, hardness values without regard to the standard deviation are used in the algorithm, especially as defining interval limits in the EKF may lead to a distortion of the prediction due to the loop nature of the algorithm. Further studies on how interval limits influence predictions are recommended.

The algorithm uses the hardness, calculates the local strength and residual hoop stresses, and runs the EKF. After strength is predicted, the plasticity level will be forecast by scanning the tensile test data. Figure 11c shows the prediction of the plasticity level.

It can be seen that compared to the measurement values derived by regression, the predictions forecast the same trends. Both local strength and residual hoop stress development is predicted in the same trend as the regression derives. Yet, outliers in the predictions made by the EKF can be seen.

Thus, the current algorithm successfully offers an estimate as well as a prediction of the local strength, residual hoop stresses and plasticity. Yet, further validation is needed. Hence, more mechanical property data in the form of tensile test and residual stress analyses need to be generated. As the current modelling of noise in the EKF is solely dependent upon the taken measurements, further analyses regarding disturbances due to the process will be conducted and modelled in the filter. Furthermore, a detailed characterization of the base material to adequately identify disturbances due to scatter bands in the mechanical properties is anticipated and will be incorporated into the soft sensor. It is also intended to implement error propagation rules into the algorithm to get an even clearer prediction of the mechanical properties. As a model for the initial soft sensor for freeform bending, the EKF has proven to be a suitable algorithm as it linearizes the nonlinear system and can be customized according to preference.

As the material, seen in Figure 7, does not have a pronounced strain hardening behavior, strength values do not offer a large range. This can lead to difficulties in deriving the plasticity level later in the algorithm as little change in the local strength suggests a larger interval of plasticity in the algorithm. Further studies, especially regarding the local plasticity level, as mentioned in Section 3.5, are intended to incorporate the plasticity level into the EKF.

The greatest advantage of utilizing a correlation scheme based on hardness is that it can be transferred to various kinds of materials. The authors investigated what inline testing equipment can be utilized as a basis for a soft sensor. The most promising approaches were either using micromagnetic Barkhausen sensors or hardness measurements. The authors decided to formulate an initial soft sensor based on hardness measurements, as this does not exclude non-ferromagnetic materials. Yet, additional investigations on the implementation of nondestructive testing results generated by Barkhausen noise sensors are conducted and will be incorporated into the soft sensor. A disadvantage of the scheme is that for a new material the regressions have to be newly determined, which does take some time to test.

## 4. Conclusions

In conclusion, this paper offers an approach for the inline prediction of residual hoop stresses, local strength and plasticity level based on UCI hardness measurements. UCI hardness was validated as a suitable inline measuring system by cross-referencing it with established HV 10 measurements. The results show that an analytic description of residual stresses, local strength and plasticity based on hardness is possible and can be implemented into an EKF for the prediction of material properties. The following insights can be gleaned:UCI hardness measurement equipment is now a validated method and offers a flexible, portable measuring solution that can be integrated into the freeform bending machine.An analytic representation of residual hoop stresses, local strength and plasticity can be derived from the hardness of the material and serves as the prerequisite for formulation of the novel correlation scheme.The novel correlation scheme can be implemented into an EKF now, enabling an inline prediction of mechanical properties during freeform bending.

It must be noted that further studies, especially regarding disturbances in the process and material, as well as error propagation, are conducted and will contribute to the design of the soft sensor. Additionally, micromagnetic Barkhausen noise sensors will be integrated into the process to obtain more information, namely on changes in microstructure during the freeform bending process. This will give further insights into the change in hardness and residual stresses during bending. Furthermore, longitudinal residual stress and radial residual stress predictions will be implemented for inline prediction.

As this is a novel approach to monitoring mechanical properties during freeform bending, no comparative studies on the algorithm can yet be given. This will be the topic of further studies, as different algorithms are planned to be implemented for the freeform bending process.

## Figures and Tables

**Figure 1 materials-14-04549-f001:**
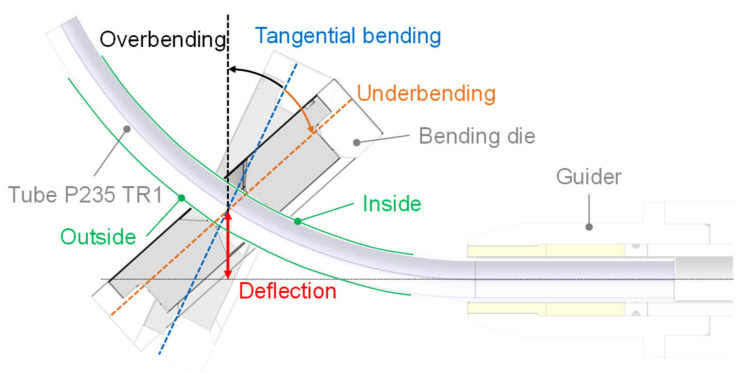
Schematic depiction of non-tangential bending (over- and underbending) for freeform bending with movable die. Adapted from Ref. [7].

**Figure 2 materials-14-04549-f002:**
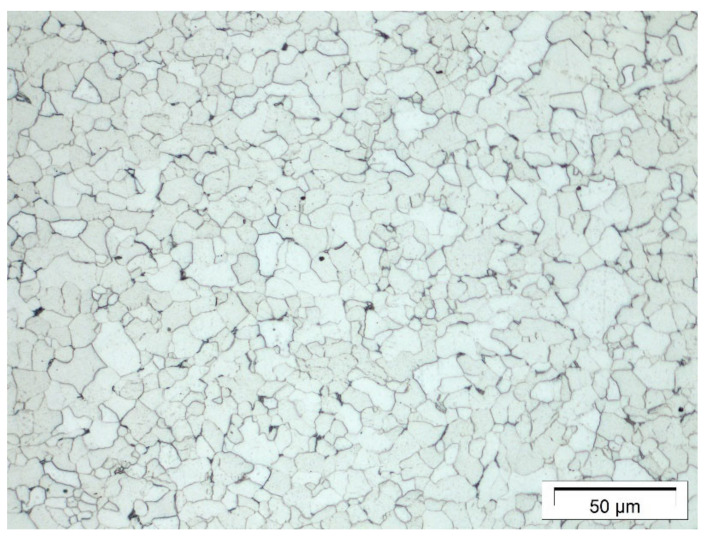
Microstructure of P235 TR 1 investigated by light optical microscope, magnification 500×.

**Figure 3 materials-14-04549-f003:**
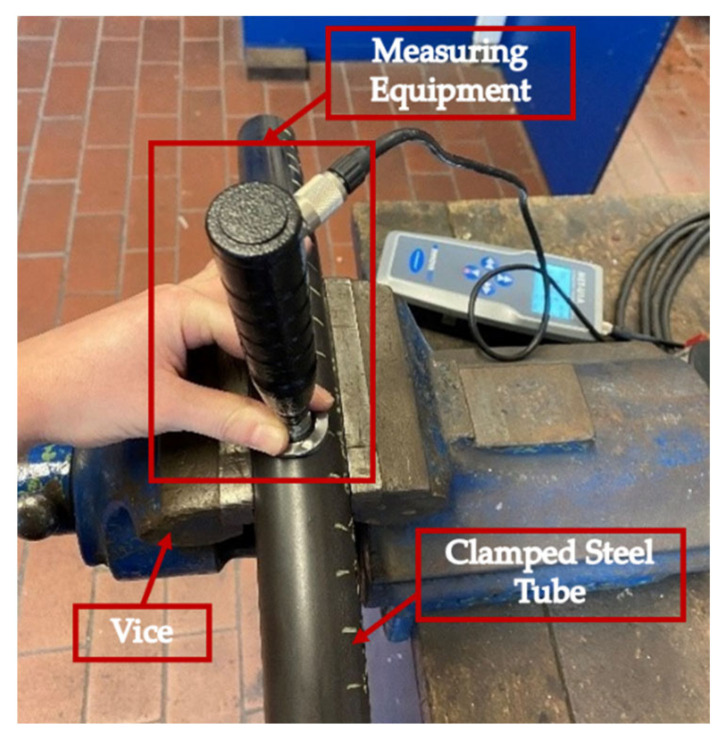
Method of conducting UCI hardness tests on bent steel tube (outside). UCI measuring equipment is used by hand to take hardness measurements of clamped steel tube.

**Figure 4 materials-14-04549-f004:**
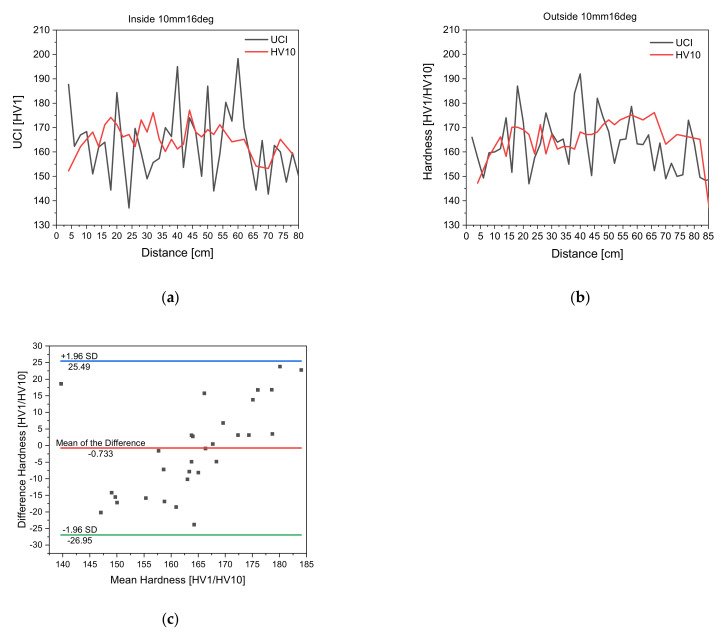
UCI hardness plots vs. HV 10 hardness plots for measurement suitability analysis; (**a**) UCI hardness vs. HV 10 measurements of non-tangentially bent tube with bending strategy 10 mm 16 deg on the inside; (**b**) UCI hardness vs. HV 10 measurements of non-tangentially bent tube with bending strategy 10 mm 16 deg on the outside; (**c**) Bland–Altman plot for concordance analysis between UCI hardness and HV 10 hardness measurements taken on 10 mm 16 deg bent tube.

**Figure 5 materials-14-04549-f005:**
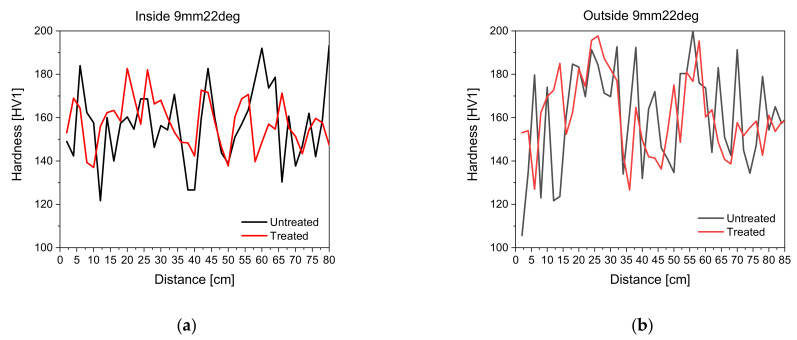
Development of UCI hardness on treated and untreated surfaces along the inside and outside of the steel tube; (**a**) UCI hardness development on untreated and treated surfaces on inside of steel tube 9 mm 22 deg; (**b**) UCI hardness development on untreated and treated surfaces on outside of steel tube 9 mm 22 deg.

**Figure 6 materials-14-04549-f006:**
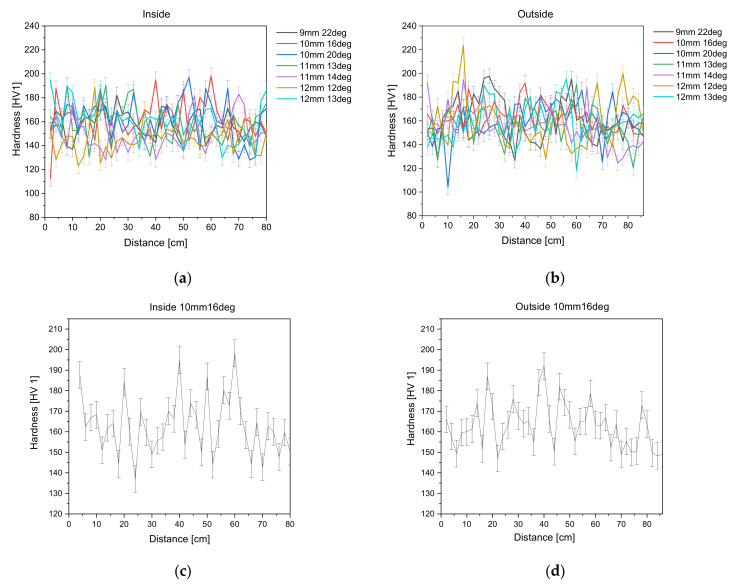
Development of HV 1 hardness total along the (**a**) inside and (**b**) outside of the non-tangentially bent tubes (treated surface). (**c**) Development of HV 1 hardness for non-tangentially bent tube 10 mm 16 deg inside (**d**) and outside.

**Figure 7 materials-14-04549-f007:**
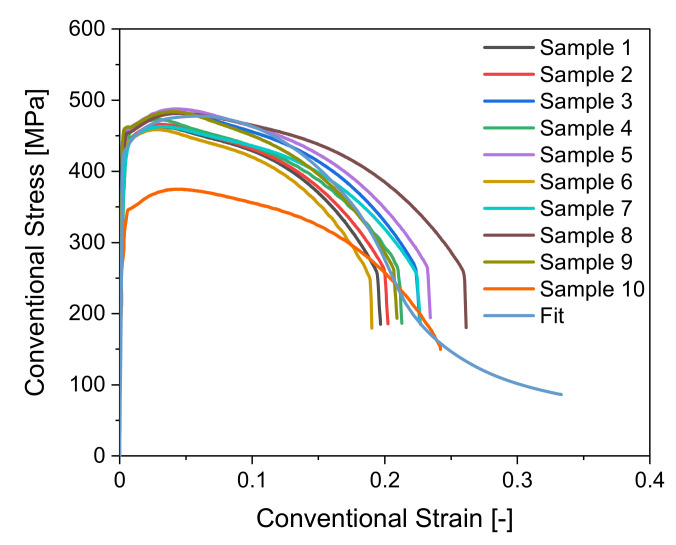
Conventional stress–strain curves with fitted curve.

**Figure 8 materials-14-04549-f008:**
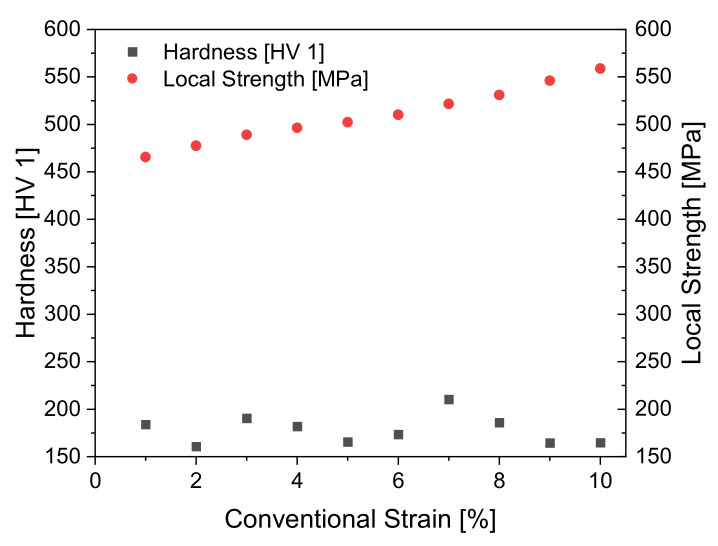
Relationship between hardness and local strength.

**Figure 9 materials-14-04549-f009:**
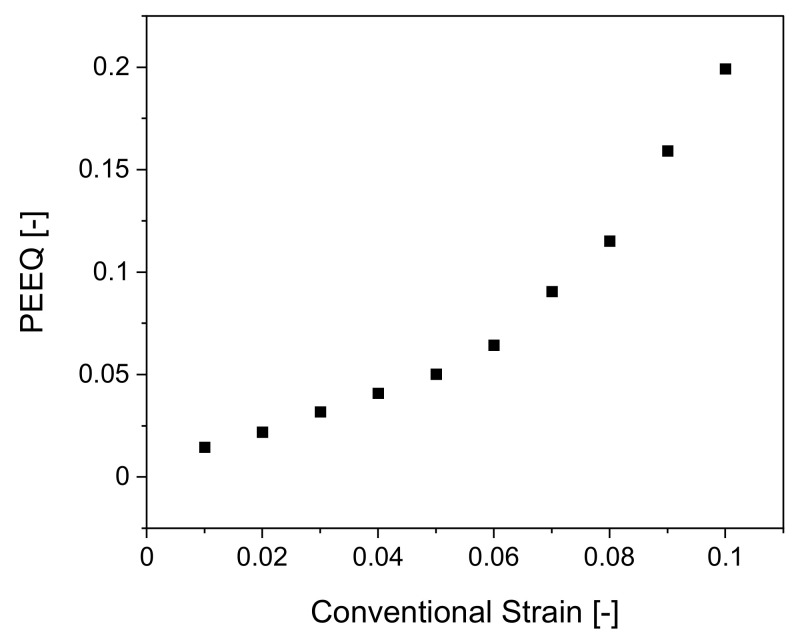
Conventional Strains from tensile tests plotted against equivalent plastic strain (PEEQ) values from simulation for derivation of true strain level introduced by bending.

**Figure 10 materials-14-04549-f010:**
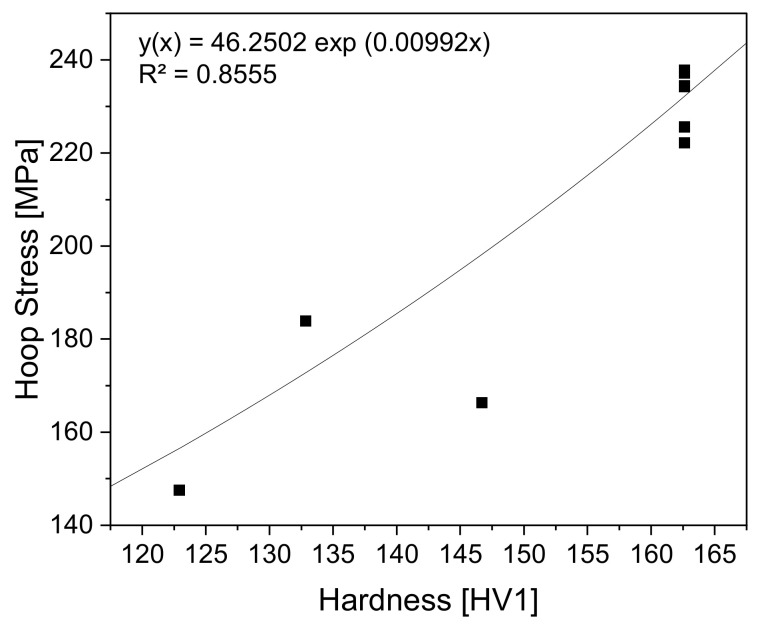
Relationship between hardness and residual hoop stresses derived from results in. Adapted from Ref. [7].

**Figure 11 materials-14-04549-f011:**
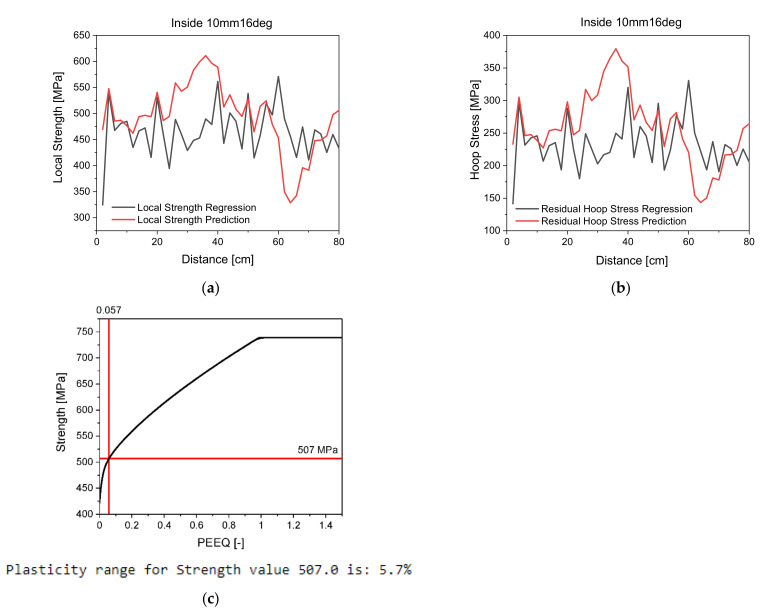
Development of mechanical properties predicted by EKF; (**a**) Local strength derivation by regression and prediction; (**b**) Residual hoop stress derivation by regression and prediction; (**c**) Derivation of plasticity level in tube through predicted strength from simulation data.

**Table 1 materials-14-04549-t001:** Chemical composition of P235 TR 1 in mass percent investigated by optical emission spectrometer.

Element	C	Si	Al	Mn	P	S	Cr	Cu	Mo	Ni
P235 TR 1	0.037	0.027	0.025	0.11	0.0048	0.0016	0.041	0.244	0.011	0.081

**Table 2 materials-14-04549-t002:** Fitting parameters for extrapolated flow curve in simulation model.

Parameter	$\alpha_{L\;}$	$b_{L}$	$A_{L}$	$n_{L}$	$k_{0,V}$	$Q_{V}$	$\beta_{V}$
Value	0.225	425	1200	0.7	420	65	62.92

**Table 3 materials-14-04549-t003:** Determination of statistical measurement uncertainty by standard deviation according to Formula (7).

**Measurement**	**1**	**2**	**3**	**4**	**5**	**6**	**7**	**8**	**9**	**10**	**s= 6.5** **HV 1**
HV1	187	191	198	196	191	184	200	191	203	192	
**Measurement**	**11**	**12**	**13**	**14**	**15**	**16**	**17**	**18**	**19**	**20**	
HV1	189	185	200	189	190	187	187	190	201	204	
**Measurement**	**21**	**22**	**23**	**24**	**25**	**26**	**27**	**28**	**29**	**30**	
HV1	186	200	192	182	194	188	180	183	185	189	

## Data Availability

Not applicable.

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
