# Peer review of "A System Identification and Implementation of a Soft Sensor for Freeform Bending"

_materials, 2021, doi:10.3390/ma14164549_

Round 1

Reviewer 1 Report

This paper investigates the soft sensor using EKF. The practical contribution is clear and the technical approach is sound. However, some comments should be further considered as a minor revision.

1) Where is the randomness from? If this is from measurement, does it obey Gaussian distribution? Is there non-linear non-Gaussian dynamics in the process?

2) EKF is a model-based design where the model has been mentioned in the algorithm 1, in particular function g() and h(). However, the details of the functions have not been given across the manuscript.  Please supply the complete model of the process in details.

3) The results have been given as the validation however no comparison is introduced. Why the authors select EKF as the filtering algorithm and if the performance can be improved if other algorithms are adopted? 

4) For non-Gaussian filtering, the following papers probably are beneficial to the extended research works in future: RBFNN-based Minimum Entropy Filtering for a Class of Stochastic Nonlinear Systems; An EKF-based performance enhancement scheme for stochastic nonlinear systems by dynamic set-point adjustment;Performance enhanced Kalman filter design for non-Gaussian stochastic systems with data-based minimum entropy optimisation, etc.

Reviewer 2 Report

Dear authors, 

Please see the attached archive for my review of your manuscript.  

Reviewer 3 Report

This work is quite interesting and some good results were reported. The model scheme was well-designed and the relevant data was systematically analyzed. Impressive amount of work. The manuscript is well-written and I don't have any specific questions on this manuscript.

Author Response

Dear Reviewer,

Thank you very much for the positive feedback on our manuscript. 
We wish you all the best and hope you stay safe in these times.

With kind regards,

The authors